

# Denoising the Denoisers: an independent evaluation of microbiome sequence error-correction approaches

Jacob T. Nearing[1], Gavin M. Douglas[1], André M. Comeau[2] and Morgan G.I. Langille[1,3]

[1] Department of Microbiology and Immunology, Dalhousie University, Halifax, Nova Scotia, Canada
[2] Integrated Microbiome Resource, Dalhousie University, Halifax, Nova Scotia, Canada
[3] Department of Pharmacology, Dalhousie University, Halifax, Nova Scotia, Canada

## ABSTRACT

High-depth sequencing of universal marker genes such as the 16S rRNA gene is a common strategy to profile microbial communities. Traditionally, sequence reads are clustered into operational taxonomic units (OTUs) at a defined identity threshold to avoid sequencing errors generating spurious taxonomic units. However, there have been numerous bioinformatic packages recently released that attempt to correct sequencing errors to determine real biological sequences at single nucleotide resolution by generating amplicon sequence variants (ASVs). As more researchers begin to use high resolution ASVs, there is a need for an in-depth and unbiased comparison of these novel "denoising" pipelines. In this study, we conduct a thorough comparison of three of the most widely-used denoising packages (DADA2, UNOISE3, and Deblur) as well as an open-reference 97% OTU clustering pipeline on mock, soil, and host-associated communities. We found from the mock community analyses that although they produced similar microbial compositions based on relative abundance, the approaches identified vastly different numbers of ASVs that significantly impact alpha diversity metrics. Our analysis on real datasets using recommended settings for each denoising pipeline also showed that the three packages were consistent in their per-sample compositions, resulting in only minor differences based on weighted UniFrac and Bray–Curtis dissimilarity. DADA2 tended to find more ASVs than the other two denoising pipelines when analyzing both the real soil data and two other host-associated datasets, suggesting that it could be better at finding rare organisms, but at the expense of possible false positives. The open-reference OTU clustering approach identified considerably more OTUs in comparison to the number of ASVs from the denoising pipelines in all datasets tested. The three denoising approaches were significantly different in their run times, with UNOISE3 running greater than 1,200 and 15 times faster than DADA2 and Deblur, respectively. Our findings indicate that, although all pipelines result in similar general community structure, the number of ASVs/OTUs and resulting alpha-diversity metrics varies considerably and should be considered when attempting to identify rare organisms from possible background noise.

Corresponding author
Morgan G.I. Langille,
morgan.langille@dal.ca

## INTRODUCTION

Microbiome studies often use an amplicon sequencing approach where a single genomic region is sequenced at a sufficient depth to provide relative abundance profiles of the microbes present in a sample. The 16S rRNA gene (16S) is usually chosen as a marker gene for sequencing of bacterial communities due to its unique structure that contains both conserved and variable regions and its presence in all known Bacteria and Archaea species. This sequencing approach is often used to avoid the high cost of shotgun metagenomic sequencing or to avoid problems with sequencing non-microbial DNA from host contamination. However, sequencing errors make it difficult to distinguish biologically real nucleotide differences in 16S sequences from sequencing artefacts. To avoid this issue sequences are often clustered into operational taxonomic units (OTUs) at a particular identity threshold (e.g., 97%) to avoid the problem of differentiating biological from technical sequence variations; however, this comes at the cost of taxonomic resolution. Recently, many new bioinformatic sequence "denoising" approaches have been developed to address this issue by attempting to correct sequencing errors thus improving taxonomic resolution. These pipelines differ in how they correct sequencing errors. DADA2 generates a parametric error model that is trained on the entire sequencing run and then applies that model to correct and collapse the sequence errors into what the authors call amplicon sequence variants (ASVs) (*Callahan et al., 2016*). This approach is advantageous as it builds unique error models for each sequencing run. Deblur aligns sequences together into "sub-OTUs" and, based on an upper error rate bound along with a constant probability of indels and the mean read error rate, removes predicted error-derived reads from neighboring sequences (*Amir et al., 2017*). Deblur employs a sample-by-sample approach which reduces both memory requirements and computational demand. UNOISE3 uses a one-pass clustering strategy that does not depend on quality scores, but rather two parameters with pre-set values that were curated by its author to generate "zero-radius OTUs" (*Edgar, 2016*). The advantage of a one-pass clustering strategy is that it saves on the computational time required to analyze the sequences in the provided study. Note that ASVs, sub-OTUs, and zero-radius OTUs are synonymous and the term ASV will be used henceforth. Denoising approaches provide improved resolution and they avoid having to make a choice between various OTU strategies which may result in differing results (*Edgar, 2017*). In addition, ASVs can be identified by their unique biological sequences instead of relying on per-study IDs, which allows for easier comparison across datasets (*Callahan, McMurdie & Holmes, 2017*).

Although there have been several bioinformatic comparisons of OTU-based approaches (*Allali et al., 2017*; *Plummer & Twin, 2015*), a thorough third-party comparison of denoising pipelines has yet to be conducted. In this paper, we compare the strengths and weaknesses of the DADA2, UNOISE3, and Deblur packages along with a comparison to an open-reference 97% OTU-based approach (*Rognes et al., 2016*), while following the recommended default settings. We assess the accuracy of these approaches using several mock communities including both bacterial and fungal amplicons. In addition, we compare the results of the four pipelines on three previously-published real human, mouse, and soil datasets.

## METHODS

### Sequence acquisition

The HMP mock community (which has even expected abundances) and the ZymoBIOMICS Microbial Community Standard (referred to as the Zymomock community) were sequenced by the Integrated Microbiome Resource at Dalhousie University using an Illumina MiSeq on separate sequencing runs, as previously described using the V4–V5 16S rRNA gene region (*Comeau, Douglas & Langille, 2017*). Reads were then uploaded to the European Nucleotide Archive (ENA) under accession number PRJEB24409. The Extreme dataset (mock-12) originally presented in the DADA2 paper and the fungal ITS1 dataset (mock-9) were retrieved from the Mockrobiota project (*Bokulich et al., 2016*). The Extreme dataset was sequenced using an Illumina MiSeq (*Callahan et al., 2016*) and the fungal mock community was sequenced using an Illumina HiSeq (*Bokulich et al., 2016*). The soil data set collected from blueberry fields (available under NCBI SRA PRJNA389786) (*Yurgel et al., 2017*) and the exercise dataset collected from stool of mice that exercised plus controls (ENA accession PRJEB18615) (*Lamoureux, Grandy & Langille, 2017*) as well as the human-associated dataset of intestinal biopsies of pediatric Crohn's disease patients plus controls (ENA accession PRJEB21933) (*Douglas et al., 2018*) were sequenced at the Integrated Microbiome Resource at Dalhousie University.

### Filtering

All sample data were filtered using the Microbiome Helper filtering scripts (*Comeau, Douglas & Langille, 2017*). In summary, primers were trimmed off all reads using Cutadapt (v 1.14) (*Martin, 2011*) and GNU Parallel (*Tange, 2011*). Primer-free sequences were then input into the dada2_filter.R script available in Microbiome Helper. This script takes in the maximum expected number of errors allowed as well as a truncation length. The HMP mock community and the Zymomock community were truncated to read lengths of 270 and 210 base pairs for the forward and reverse reads respectively to remove low quality bases at the end of the reads. The shorter length for the reverse reads is a result of their lower overall quality compared to the forward reads. Note that for DADA2, and Deblur to work properly reads need to be of the same length. The single-end reads from the Extreme mock community and the fungal mock community were truncated to 80 base-pairs. The three real datasets: soil, mouse, and human-associated, were truncated to 270 and 210 base-pairs for the forward and reverse reads, respectively. The number of expected errors allowed were defined as three different filtering stringencies: 5 (low), 3 (medium), and 1 (high).

### DADA2 pipeline

The DADA2 pipeline was run using Microbiome Helper scripts, which wraps the core algorithms of the DADA2 pipeline (*Callahan et al., 2016*). Filtered reads were input into the wrapper script dada2_inference.R which runs the DADA2 inference algorithm. Once ASVs are determined, they are passed into DADA2's chimera-checking algorithm which was run using the wrapper script dada2_chimera_taxa.R to screen out chimeric sequences.

The output objects containing ASV sequences and abundances counts were then converted into BIOM table format using convert_dada2_out.R. All DADA2 wrapper scripts were run with default settings.

### UNOISE3 pipeline

Filtered reads were input into USEARCH's (v 10) (*Edgar, 2010*) fastq_mergepairs command if they were paired-end reads or concatenated together into one FASTQ if they were single-end reads. Next, the merged FASTQ file was converted into a FASTA file using the Microbiome Helper script run_fastq_to_fasta.pl and then used as input for USEARCH's fastx_uniques command which generated a FASTA containing all the unique sequences and their abundance. Finally, the FASTA containing unique sequences was used as input into USEARCH's unoise3 (*Edgar, 2016*) command generating a BIOM table and representative ASVs that were used in subsequent analyses. All USEARCH scripts were run with default settings.

### Deblur pipeline

Paired-end filtered reads were stitched together using the Microbiome Helper wrapper script run_pear.pl which wraps the program PEAR (v 0.9.10) (*Zhang et al., 2014*). This step was skipped for filtered single-end reads. Next, reads were renamed to match a format that was compatible with QIIME2 (*Caporaso et al., 2010*) and converted into a QIIME2 artifact. Samples were then run through QIIME2's built-in deblur command using the 16S setting, which uses Greengenes 13_8 (*DeSantis et al., 2006*) for positive filtering. A separate analysis to test the effect of positive filtering was conducted by using the non-positive filtered output files ("all.biom" and "all.seqs.fa") from the stand alone version of Deblur. Fungal reads were run using the "other" setting and the UNITE 10.10.2017 database (*Kõljalg et al., 2013*). Finally, the representative ASV sequences and a BIOM table were exported from the QIIME2 artifact. All other Deblur scripts were run with default settings.

### Open reference OTU pipeline

Filtered reads were stitched and imported into a QIIME2 artifact as previously described in the Deblur pipeline. The imported sequences were then dereplicated using the QIIME2 VSEARCH plugin. Dereplicated sequences were filtered for chimeras using the QIIME2 VSEARCH UCHIME reference-based chimera filtering (*Edgar et al., 2011*) using the Greengenes13_8 97% OTU database for reference. Chimera-checked sequences were then clustered at 97% in an open-reference fashion using the QIIME2 VSEARCH cluster-features-open-reference plugin. The Greengenes 13_8 97% OTU database was used as the reference database during open-reference OTU picking. Finally, singleton OTUs were removed from the OTU table and the table was exported in BIOM format along with its representative sequences in FASTA format.

### Run time and memory analysis

The soil dataset was filtered using the low-stringency filter and then individual samples were rarefied to either 5,000, 10,000, 20,000 or 30,000 reads per sample. The different read-depth sets were then run through the three denoising pipelines and user time and maximum memory usage was determined using the GNU time (v 1.7) command.

## ASV and OTU analysis of mock communities

ASVs and OTUs were compared against the expected sequences provided with each of the mock communities. This comparison was done using the command-line BLASTN (v 2.7.10) (*Altschul et al., 1990*) tool against the expected sequences from each community. This allowed us to determine the number of full length 100% matches and 97% matches. All ASVs and OTUs that did not match these criteria were then compared against the SILVA 16S rRNA gene database (v 128) (*Pruesse et al., 2007*) or the UNITE 10.10.2017 database to find all full length 100% and 97% matches. Any ASVs or OTUs that did not match these databases were then labeled as "Unmatched". The number of unique expected sequences for each mock community was determined by slicing out the amplified regions using a custom Python (v 3.6.1) script (slice_amplified_region.py, available on GitHub) from the expected sequences of each mock community and then finding the number of unique sequences from this output using USEARCH10's fastx_uniques command. In addition to analysis done with no abundance filtering, we also compared how filtering of low abundance ASVs/OTUs affected the type and amount of ASVs/OTUs called by each pipeline. This was done by applying a 0.1% minimum abundance filter to each approach and dataset except for the Extreme mock community where a 0.0004% minimum abundance filter was applied. The 0.1% minimum abundance filter was chosen based on the known 0.1% bleed through between Illumina MiSeq runs. A lower minimum abundance filter of 0.0004% was chosen for the Extreme mock community since some of the expected sequences had a lower expected abundance than 0.1%. Histograms with sequence identity to the expected sequences were generated for each pipeline and each mock community except for the fungal mock community due to the community missing some expected sequences from its expected sequence list. All sequences that showed less than 75% identity were binned together.

## Abundance data analysis of mock communities

For the HMP, Zymomock and Extreme datasets all ASVs/OTUs that matched at 97% identity or greater with the provided expected sequences based on a BLASTN search were added to the abundance of the corresponding expected taxa. Stacked bar charts of expected taxa relative abundances were created using the ggplot2 (v 2.2.1) (*Wickham, 2009*) R (v 3.4.3) (*R Development Core Team, 2008*) package and the cowplot (v 0.9.2) R package (see Data Deposition for exact scripts).

Due to the incomplete nature of the expected sequences for the fungal mock community, UNITE database hits at 97% or greater to an expected species within the community were considered as expected ASVs/OTUs. All other ASVs/OTUs were classified as "Non-Reference" hits.

## Analysis of real datasets

Sequences from the three real datasets; soil, human-associated, and exercise were filtered using medium stringencies (allowing up to three expected errors per sequence) for each package and rarified to 5,215, 3,000 and 4,259 reads, respectively. These rarefaction levels were chosen as they were the lowest read count above 2,000, after the DADA2 pipeline was

**Table 1 Qualitative comparison of DADA2, Deblur, and UNOISE3.**

| Pipeline | Implemented in | Open source | [a]Pooled sampling | [b]Positive filtering | Version tested | GUI via QIIME2 | Publication date |
|---|---|---|---|---|---|---|---|
| DADA2 | R | Yes | Yes | No | 1.6 | Yes | April 13 2016 |
| Deblur | Python | Yes | No | Yes | 1.0.2 | Yes | March 7, 2017 |
| UNOISE3 | C ++ | No | Yes | No | 3 | No | Oct 15, 2016 |

**Notes.**
[a]When all sequences from all samples are denoised at the same time (in contrast to running each sample separately).
[b]Compares resulting ASVs to a database (Greengenes for Deblur) and discards reads if they do not match a certain identity threshold (88% for Deblur).

complete. ASV abundance tables output by all approaches were combined into a single table where each biological sample was represented multiple times (once for each pipeline). ASVs/OTUs not called by a specific pipeline were given an abundance of zero in their column (e.g., ASVs only called by Deblur for sampleA were given zero abundances in the columns for DADA2's and UNOISE3's outputs of sampleA). Representative sequences were aligned against the Greengenes 13_5 99% OTU aligned sequences and placed on the Greengenes 13_5 99% OTU tree using SEPP (*Mirarab, Nguyen & Warnow, 2011*). Weighted UniFrac and unweighted UniFrac distances at the ASV/OTU level were then generated using the QIIME1 beta_diversity.py command and principal coordinates were generated using the QIIME1 prinicpal_cordinates.py command.

The Bray–Curtis distance matrix at the genus level was generated by assigning taxonomy to the resulting ASV/OTUs from each package using the RDP classifier (*Cole et al., 2014*) with the assignTaxonomy function available in the DADA2 package and the rdp_train_set_16 database. Distances were then generated using the summarize_taxa.py and beta_diversity.py commands in QIIME1. Ordination was generated using the metaMDS function in the vegan R package (*Oksanen et al., 2018*).

Mantel correlations between each distance matrix for each pipeline and each beta-diversity metric (weighted/unweighted UniFrac and Bray–Curtis) were generated using the vegan R package. The number of observed OTUs/ASVs per sample for each real dataset were generated using the rarified combined OTU/ASV tables and the alpha_diversity.py command in QIIME1 using observed OTUs as the metric option.

## RESULTS

### Differences in methodology and availability between different denoising pipelines

There are important aspects other than accuracy that need to be considered when determining which denoising package a researcher should use for their project. Both DADA2 and UNOISE3 are suggested to be run in a pooled-sample workflow, where all sequences are pooled together during the denoising process (Table 1). This allows them to better account for batch errors across multi-run experiments. Deblur, on the other hand, runs its denoising process sample-by-sample. This approach helps lower Deblur's computational requirements, but at the cost of reducing its ability to correct multi-run batch effects. Both DADA2 and Deblur are open source projects, whereas UNOISE3 is a

closed-source project which has a free 32-bit academic version with a 4 Gb memory cap and a full 64-bit version that costs between $885–1,485 USD (Table 1). Another major difference is that the built-in Deblur function in QIIME2 has a positive filtering process. This default setting causes Deblur to discard reads that do not reach a length-scaled bit score and an *e*-value threshold to any sequence in the 88% representative sequences Greengenes database. Note the default database can be changed using the "other" version of the Deblur plugin in QIIME2, an important feature when working with fungal or eukaryotic data. It is also important to note that the stand-alone version of Deblur outputs both positively filtered and non-filtered results by default, unlike the QIIME2 plugin which, only outputs the positively filtered results. Currently, the functionality of both DADA2 and Deblur can be accessed through a graphical user interface as plugins in QIIME2 Studio, whereas UNOISE3 does not support a graphical user interface (Table 1).

## Total number of ASVs/OTUs vary across approaches in mock communities

We processed four different mock communities with the DADA2, UNOISE3, and Deblur denoising pipelines as well as an open-reference 97% OTU clustering pipeline to compare the resulting ASVs/OTUs from each approach. Focusing on the number of called ASVs from each denoising pipeline, we found no approach consistently called more ASVs. DADA2 called the most ASVs in two mock communities (HMP: 42, Extreme: 78) and UNOISE3 called the most ASVs in the other two mock communities (Fungal: 38, Zymomock: 43) under medium stringency filtering (Fig. 1, Table S1). Overall, open-reference OTU clustering output the most ASVs/OTUs for all four mock communities (HMP: 453, Extreme: 8891, Fungal: 96, Zymomock: 294). None of the approaches were capable at outputting all expected sequences at 100% identity in any of the mock communities that were processed and, in all datasets, at least one denoising pipeline output more ASVs than expected sequences. All four approaches output at least one ASV/OTU at 97% or greater identity from all organisms in the HMP mock community and the Zymomock community (Tables S2, S3). DADA2 output nine more ASVs with 97% or greater identity matches to expected sequences in the Extreme dataset (read depth of 2 million reads) than the other two denoising pipelines (Table S4). Six of the nine taxa that DADA2 called and the other pipelines did not call, had expected relative abundances of only 0.000427% (Table S4). Of the other three taxa not found by Deblur or UNOISE3, one was at an expected abundance of 0.00427% and two had an expected abundance of 0.0427% (Table S4). Interestingly, open reference OTU clustering called OTUs corresponding to these nine expected taxa along with two more expected taxa that DADA2 missed, both of which were in the lowest possible expected abundance range (0.000427%). One organism was missed by all pipelines in the Extreme community (*C. methylpentusum*), which was also in the lowest possible expected abundance range. Sequence identity histograms were constructed for the HMP, Zymomock and Extreme communities (Figs. S1–S3). In the HMP community, no ASV was found to have below 80% identity by the denoising pipelines, but open reference OTU clustering found 21 OTUs below 80% identity. Furthermore, the OTU pipeline called 324 OTUs below 99% identity whereas all denoising approaches called less than 15 ASVs

Peer

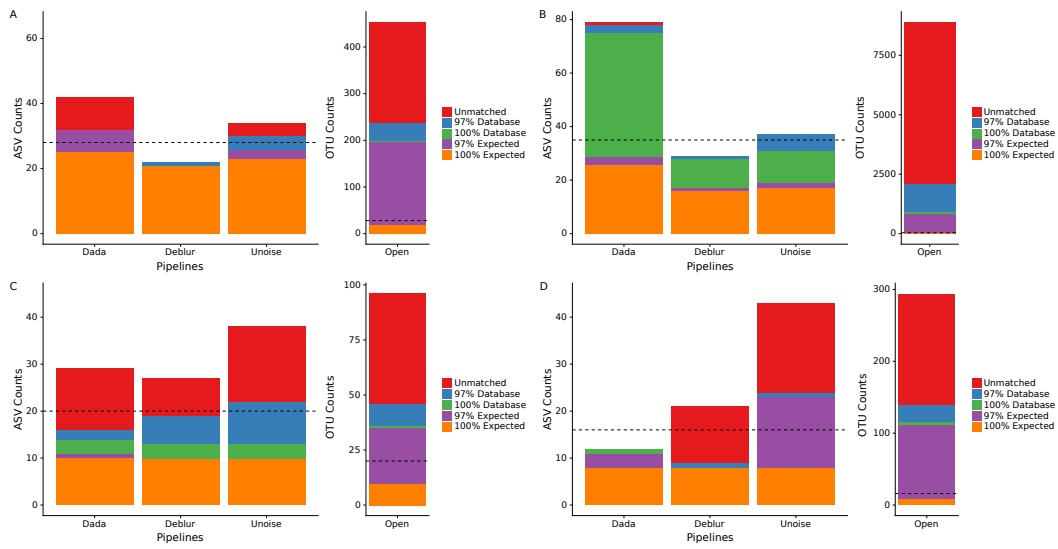

**Figure 1** **Total number of ASVs/OTUs identified by each sequence processing method for four different mock communities.** Amplicon sequence variants/Operational taxonomic units (ASVs/OTUs) were compared to a database of full-length amplicon sequences for just the microbes supposedly in the community ("Expected") and against the full SILVA or ITS databases ("Database") using BLASTN at 97% and 100% identity cut-offs. "Unmatched" sequences did not match an expected sequence or the SILVA/ITS databases at 97% identity or greater. Dotted lines indicate the total number of ASVs/OTUs expected, accounting for 16S copy variation within genomes. Note that the *y*-axis for open-reference OTU clustering is different than the *y*-axis on the denoising methods. (A) Human Microbiome Project equal abundance mock community; (B) Extreme dataset; (C) Fungal ITS1 mock community; (D) Zymomock community.

(Fig. S1). In the Extreme community, most ASVs were found in the 99–100% identity range whereas the majority of OTUs found by open-reference OTU clustering were in the 95–97% identity range (Fig. S2).

Due to the expected sequences for the fungal community being incomplete, we included any UNITE database hits as expected sequences if the matching sequence was from a species that was included in the fungal community. This resulted in almost all the fungi present in the community to be found by all four pipelines except for *Penicillium allii* and *Penicillium commune*. Open-reference OTU clustering did find *P. allii*, although at an extremely low abundance of 0.004% compared to its expected abundance of 6.25%.

Given that some of the above potential spurious ASVs would be removed by sequence bleed-through (*Illumina, 2017*) or low-abundance filters in typical workflows, we applied an abundance cutoff filter of 0.1% to each mock community except for the Extreme community, where an abundance filter of 0.0004% was applied (Fig. S4). This lower abundance filter was chosen due to some expected sequences being in abundance of 0.000427%. Application of the abundance filter to the HMP community resulted in all 10 unmatched ASVs (those that did not match either the expected or SILVA by 97% or greater) called by DADA2 to be discarded, but none of the four unmatched reads in UNOISE3 to be discarded (Fig. S4). A similar phenomenon was seen in the Zymomock community with all 12 of Deblur's unmatched reads being discarded (along with one database hit) and

UNOISE3 only discarding one of 19 unmatched reads it called. The largest effect that the application of the filters had was the removal of a significant amount of OTUs found in each mock community by the open-reference OTU pipeline. The most drastic change was seen in the Extreme community that started with 8,891 OTUs and was reduced to 1,248 OTUs (Fig. S4). In the fungal community the number of OTUs found by open-reference OTU clustering (27) became less than the number of ASVs found by UNOISE3 (36).

To determine how read quality filtering affects the number of ASVs called by each pipeline, we ran each denoising pipelines using two additional quality filtering stringencies, low and high (see 'Methods'). The different quality filter stringencies used made only small impacts on the numbers of ASVs called by each pipeline for the HMP, Extreme and fungal datasets (Table S1). A difference of six ASVs was the largest between the high and medium stringencies on the HMP community and was output by UNOISE3 (Table S1). In the Zymomock community, the number of ASVs called by DADA2 only varied by one for all three stringencies, but Deblur varied by as much as 12 ASVs and UNOISE3 varied by as much as 16 ASVs being called between the high and medium filter stringencies (Table S1).

## Denoising pipelines are consistent in determining mock community composition

Despite the different ASV counts between each pipeline in the mock community, the relative abundances of the expected taxa are strikingly similar (Fig. 2). In both the HMP and Zymomock communities, only a small portion of ASVs called by DADA2 and Deblur did not match the expected sequences by 97% identity or greater. In contrast, UNOISE3 identified multiple (eight in HMP, 20 in Zymomock) sequences that summed together to make up 2.9% and 4.8% of the relative abundance in the HMP and Zymomock communities, respectively (Tables S1, S2, S3). Open-reference OTU clustering found 6783 non-reference OTUs (Table S1) in the Extreme community that summed together to make up 2.6% of the community, whereas the denoising approaches all found non-reference abundances less than or equal to 0.3% (Table S4). None of the approaches were good at distinguishing the proper abundances of the two *Parabacteroides distasonis* strains with denoising pipelines finding similar proportions of both strains and the open-reference OTU pipeline finding dominance of the 13,400 strain and not the 13,401 strain (Fig. 2B, Table S4). None of the approaches performed well at matching the expected abundance of the Zymomock community or the fungal community (Fig. 2). All three denoising pipelines called over-abundances of *Lactobacillus fermentum* in the Zymomock community, with Deblur calling the most (44.5%) and UNOISE3 calling the least (37.6%) (Table S3). Similarly, all approaches called non-reference hits in greater than 9% abundance in the fungal community (Table S5), with UNOISE3 calling the most non-reference hits (12.4%) and Deblur calling the least (9.8%). Due to all four pipelines producing similar proportions of non-reference hits in the fungal community, this could suggest that either the mock compositions are not in the expected proportions, the four pipelines are similarly biased, or that an upstream process during sequencing caused the introduction of unexpected sequences.

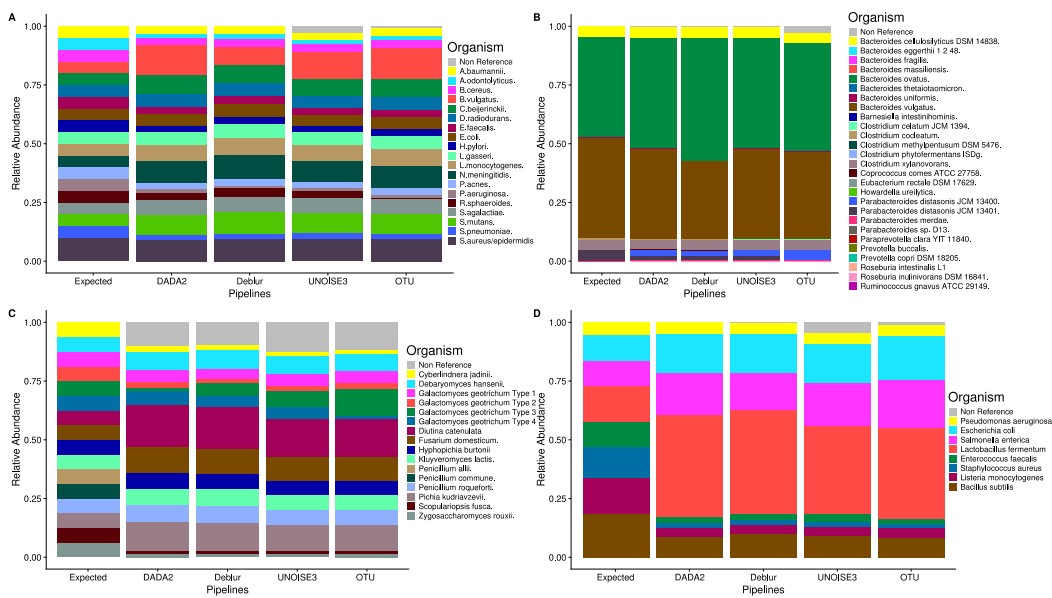

**Figure 2   Relative abundances of taxa generated by each sequence processing method for four different mock communities.** All ASVs/OTUs that matched with expected sequences at 97% or greater identity were assigned taxonomy using a BLASTN search against the expected sequences provided for the Extreme, Human Microbiome Project, and Zymomock mock communities. All ASVs/OTUs that matched an expected species with 97% or greater identity to the UNITE database were classified as expected sequences in the fungal mock community. Non-reference refers to the abundance of ASVs/OTUs that did not match expected sequences with 97% or greater identity. (A) Human Microbiome Project equal abundance mock community; (B) Extreme dataset—it is important to note that some organisms are not displayed in this figure due to their very low abundances; (C) Fungal ITS1 mock community; (D) Zymomock community.

## Weighted beta-diversity results from different approaches are indistinguishable in real soil and host-associated communities

After comparing each pipeline using mock communities, we next wanted to investigate how comparable the results between pipelines were for real 16S datasets. We compared the pipelines on a soil dataset (soil) due to its high diversity (*Fierer & Jackson, 2006*) a mouse exercise stool dataset (exercise) and a gut biopsy sample dataset from Crohn's Disease patients plus controls (human-associated). The intra-sample distances based on weighted UniFrac measurements were comparable between each approach among all three datasets (Fig. 3). In general, we found the intra-samples distances to be small, ranging between median values of ∼0.09 and ∼0.15. There were no consistent differences seen between all three datasets. Looking at the soil dataset, we found that all three denoising pipelines had similarly small intra-sample distances (∼0.11) (Fig. 3A), while the open-reference OTU clustering pipeline was slightly further away from UNOISE3 and Deblur (∼0.13). Despite this observation, it is important to note that this difference is still relatively small. Furthermore, when each sample was plotted onto a PCoA we found that samples tended to group by biological origin rather than the approach used to process the sequences, suggesting that a relatively similar PCoA plot would be generated by each pipeline (Figs. 3D–3F). The Mantel correlations between each weighted UniFrac distance
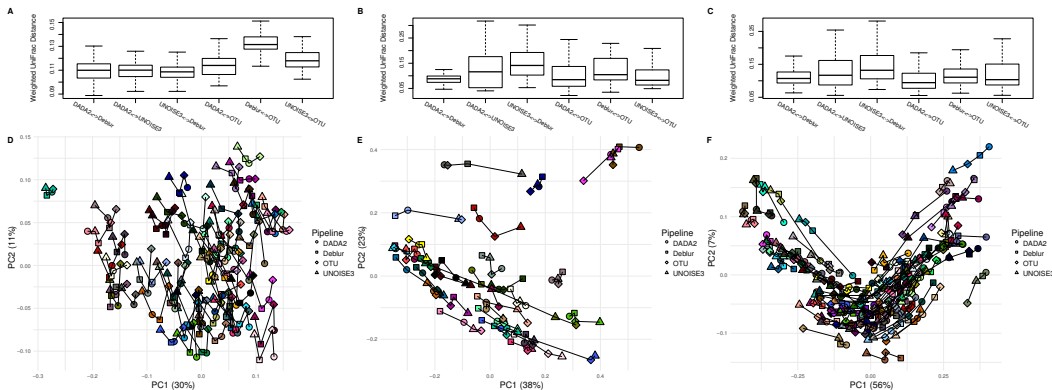

**Figure 3  Weighted UniFrac intra-sample distances between sequence processing methods based on three real datasets.** (A–C) The weighted UniFrac distances between the same biological samples based on ASVs/OTUs outputted by each of the different sequence processing methods on the soil, human associated and exercise datasets, respectively. (D–F) Principal coordinates analysis of the weighted UniFrac distances of all the samples in the soil, human associated, and Exercise datasets, respectively. The four different sample profiles generated for each biological sample are colour-coded and are joined by an interconnecting line.

matrix were all highly correlated (correlation values ranging from 0.764 to 0.975), which suggests similar weighted UniFrac profiles between each approach (Tables S6–S8).

Looking at Bray–Curtis dissimilarity, another metric that takes in abundance information, we found that intra-sample distances based on genus-level assignment were similar among different approaches (Fig. S5). In general, the intra-sample distances were relatively low, with median values ranging from ~0.04 to ~0.20, indicating a high amount of agreement between approaches. Interestingly, we found that the intra-sample distances seemed to increase across datasets based on how diverse the samples within them were (Fig. S5). In two of the three datasets (soil and BISCUIT), we found that DADA2 and UNOISE3 tended to be closer together in distance than Deblur was to DADA2 or UNOISE3, indicating a slightly higher amount of agreement between DADA2 and UNOISE3 at the genus level. Again, these differences were relatively small and would have minimal impacts on biological results obtained from them. Plotting the Bray-Curtis dissimilarity matrices onto an NMDS plot resulted in similar findings as weighted UniFrac, with samples grouping by biological origin rather than the pipeline used to process them. Furthermore, we found that the Bray-Curtis dissimilarity matrices were extremely well correlated with each other, ranging in values between 0.956-0.995 (Figs. S6–S8).

We next compared unweighted UniFrac distances, which is a metric that considers the presence or absence of ASVs/OTUs and their phylogenetic distance. We found that the median intra-sample distance between each pipeline was much greater, ranging between ~0.40 to ~0.79 (Fig. S6). Similar to our results focused on Bray-Curtis dissimilarity, we also found that the median intra-sample distance between samples tended to increase with sample diversity (0.72–0.79 for soil; 0.40–0.55 for human-associated; 0.60–0.70 for exercise). The PCoA plot based on these distances resulted in samples grouping together
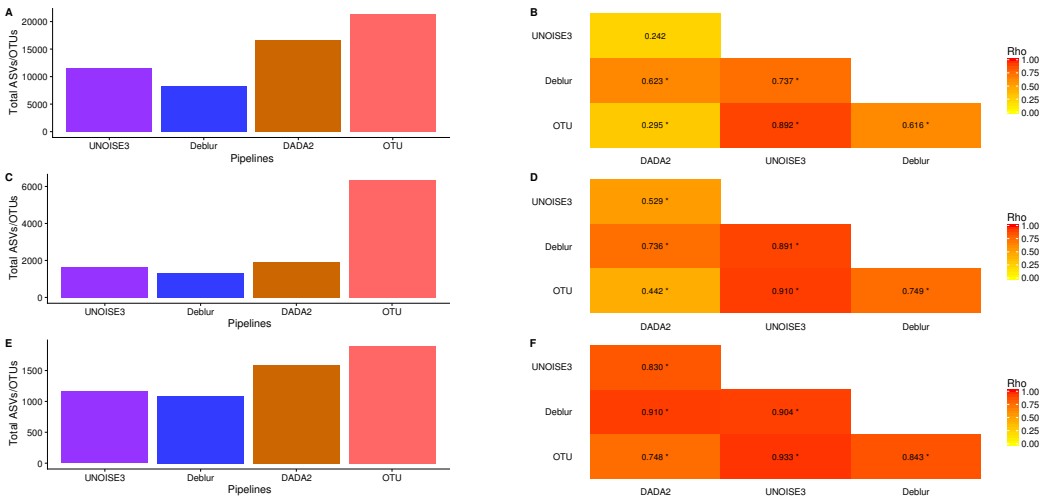

**Figure 4** **Total number of ASVs/OTUs called by each processing method and the per-sample observed ASV/OTUs correlation between each sequence processing method.** (A, C, E) The total numbers of ASVs/OTUs determined by each method on the soil, exercise, and human associated datasets, respectively. (B, D, F) Heatmaps of the Spearman correlations between the numbers of observed ASVs/OTUs per sample between different sequence processing methods. Significant $p$-values ($p < 0.05$) are indicated by *.

based on pipeline, rather than biological origin, indicating large differences between the different approaches (Fig. S6).

To look at how much of a difference the positive filtering process had on the profiles generated by Deblur and to see if the slight difference in Bray-Curtis dissimilarity among the denoising approaches was caused by this filter, we ran the stand-alone version of Deblur and examined the non-positive filtered results. We found that the intra-sample distances based on any of the three metrics tested (weighted/unweighted UniFrac and Bray-Curtis dissimilarity) between any of the pipelines did not vary (Fig. S7).

## Alpha-diversity metrics vary between denoising pipelines

We next investigated how the number of ASVs/OTUs called in a real dataset differed between processing pipelines. In the soil dataset, DADA2 called 16609 ASVs, UNOISE3 called 11613 ASVs, Deblur called 8270 ASVs and open-reference OTU clustering called 21297 OTUs (Fig. 4A). Across all datasets, we found that DADA2 called the most ASVs among the denoising pipelines and that open-reference OTU clustering called the most ASVs/OTUs overall (Figs. 4A, 4C, 4E). On average, DADA2 called 727 more ASVs than Deblur and 532 more ASVs than UNOISE3 while open-reference OTU clustering called 3135 more OTUs/ASVs than DADA2. In all datasets, Deblur called the least amount of ASVs/OTUs. Looking at the number of OTUs called per sample, we found that DADA2 did not correlate well with UNOISE3 or open-reference OTU clustering and in the soil dataset the correlation between DADA2 and UNOISE3 was found to not be significant ($p = 0.054$) (Figs. 4B, 4D, 4F). This is a concerning result as it indicated the possibility of different biological results based on the pipeline that was chosen to process the data.
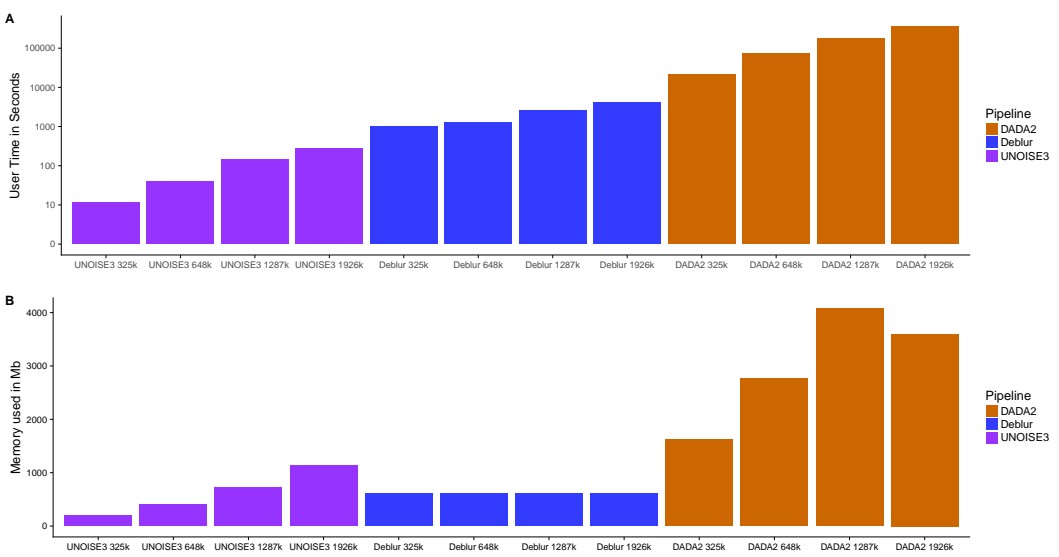

**Figure 5** **Run time and memory usage of each denoising pipeline on a dataset of varying size.** The time in seconds (A) and memory in megabytes (B) to run varying amounts of reads through the three different denoising methods. Note time is on a log_10 scale.

One interesting result was that, despite DADA2 calling the most total ASVs among the denoising pipelines, it called the least amount per sample (Figs. S8–S10) and, among denoising pipelines, UNOISE3 tended to call the most (Figs. S8–S10). However, open-reference OTU clustering called the most OTUs/ASVs overall per sample among all approaches that were tested with the exception of the soil dataset.

## Computational requirements are vastly different across denoising pipelines

Knowing that all three of these pipelines resulted in similar relative abundance profiles on mock communities and small intra-sample distances on real 16S communities, we next investigated how the run time and memory usage differed between the denoising approaches. We found that UNOISE3 (4.6 min) was 1,273 times faster than DADA2 (5,834 min) and 15 times faster than Deblur (69.3 min) at a total read count of 1,926,000 reads evenly distributed across 103 samples (Fig. 5A). Run times for all pipelines increased as the number of reads per sample increased. Deblur used a static amount of memory (611 Mb) as reads per sample increased, whereas in general the other two pipelines increased in memory usage as the number of reads per sample increased, with the exception of DADA2 run at 1,926,000 reads (Fig. 5B). Deblur used the smallest amount of memory at the maximum read count of 1,926,000 reads. We found that DADA2 had the highest amount of memory usage (4,071 Mb at 1,287,000 reads) among the three pipelines. Interestingly, this usage was more than the amount used at the maximum read count (3,600 Mb). In addition, none of the runs exceeded the 4 Gb memory cap on the 32-bit free academic version of USEARCH10.

## DISCUSSION

Despite the differences between the underlying algorithms, all four approaches were comparable based on weighted UniFrac and Bray-Curtis distances. However, the tools varied greatly when looking at unweighted or raw number of ASVs/OTUs found. It should be noted that all denoising pipelines were run using their recommended settings, limiting our comparison to only looking at the default settings for each pipeline. Adjusting of parameters within each sequence analysis pipeline would allow users to tailor towards more sensitivity and specificity as desired. Furthermore, each package uses individual chimera filtering methods which may affect the overall accuracy of the entire pipeline.

During mock community data processing, no denoising approach consistently called more ASVs than another, however open-reference OTU clustering found more OTUs than any of the three denoising approaches before abundance filtering (Fig. 1). In many cases, the number of OTUs found was vastly greater than the number of expected sequences, which is consistent with previous literature reporting that OTU clustering tends to exaggerate the number of unique organisms found within a sample (*Edgar, 2017*). No pipeline was able to call all expected sequences for any of the mock communities at 100% identity (Fig. 1), which indicates that some may not have been present or that preparation steps show large bias towards specific 16S rRNA gene variants within an organism. Each sequence processing pipeline was able to detect every organism in the HMP community (note *S. aureus* and *S. epidermidis* are collapsed together as they have the same sequenced region) and the Zymomock community. However, one odd result was that observed abundances within the HMP community were comparable to the expected abundances within the HMP community, but not the Zymomock community (Figs. 2A, 2D). This again most likely indicates bias during library preparation but could also be due to similar bias among pipelines. In the Extreme dataset, all denoising pipelines missed *P. buccalis*, *C. methylpentusum* and *P. sp._D13* (Fig. 2B, Table S4). All three of these organisms had very low expected abundances (less than 0.00427%) which may explain why they were difficult to detect (Table S4). We do not believe it was an issue with sequence depth as the single sample in the mock community was sequenced at a depth of 2 million reads and open-reference OTU clustering only missed one organism (*C. methylpentusum*). Deblur and UNOISE3 both failed to detect nine of the 27 expected taxa in the Extreme dataset at 97% identity, which were all detected by DADA2 and open-reference OTU clustering. Again, these nine organisms were at very low abundances (less than 0.05%) (Table S4). As mentioned above, open-reference OTU clustering did find two more expected taxa than DADA2, however, this came at the cost of significantly increased numbers of non-reference sequences being detected. Differences in detection between DADA2 and the other two denoising pipelines suggests that it is better at detecting organisms that are very rare without the cost of finding significantly more non-reference hits that plagues the open-reference OTU clustering pipeline. Whether finding rare organisms is truly advantageous is debatable, as many of these low-abundance organisms would be removed by typical filtering cut-offs and/or contribute little to weighted beta-diversity metrics such as UniFrac or Bray-Curtis dissimilarity.

In all mock datasets, a large number of Unmatched OTUs were found by open-reference OTU clustering and, in some of the datasets, different denoising pipelines also found a relatively large amount of Unmatched ASVs (Fig. 1). This could have been due to errors in sequencing or mutations during PCR amplification of the 16S rRNA gene as most sequences did show over 90% identity to the expected sequences, indicating that they originated from a 16S rRNA gene (Figs. S1–S3). It is also possible that many of these sequences are contaminants from the environment, while others may have been from chimeras that were not detected by each pipelines' individual chimera-checking algorithms. We also found a high proportion of matches to the reference database but not the expected sequences. There are several reasons why this could have occurred. Some of these sequences may be contaminants introduced during mock community preparation or sequencing as they match with 100% identity to a sequence within the Greengenes database indicating that they are a real 16S rRNA gene sequence. However, due to different pipelines identifying different numbers of these unexpected sequences, it could also suggest that some of these contaminants are likely either chimeric sequences that can be found within the Greengenes database, or sequencing errors/mutations during PCR amplification that could not be corrected for by the respective pipelines. Unsurprisingly, the highest sequence depth mock community (Extreme) had the most Unmatched OTUs (Fig. 1), most likely due to the increased absolute abundance of sequencing errors and the increased likelihood of finding minor contaminants. Furthermore, DADA2 found a significantly greater amount of sequences under or at 75% sequence identity, when compared to the other denoising pipelines, highlighting the trade-off that it makes to find lower-abundance ASVs (Fig. S2).

In all cases, open-reference OTU clustering had greatly increased numbers of OTUs found when compared to ASVs found by denoising pipelines (Fig. 1). To try and address this issue, we applied an abundance filter of 0.1% minimum abundance (except for the Extreme community 0.0004% minimum abundance) to all four different pipelines over all the datasets (Fig. S4). The largest effect that this filter had was the reduction of the number of OTUs called by open-reference OTU clustering. These significant changes in the number of OTUs called by open-reference OTU clustering highlight the importance of removing low-abundance OTUs during analysis and demonstrate the stability of denoising approaches after abundance filtering. The filter cutoff did have an impact on the number of Unmatched ASVs called by DADA2 in the HMP mock community (Fig. S4A) and the Unmatched ASVs called by Deblur in the Zymomock community (Fig. S4E), but had little effect on the number of ASVs called by UNOISE3 on these communities, indicating UNOISE3's tendency to call ASVs of higher abundance. No differences were seen in the Extreme community on ASV output by denoising approaches, most likely due to the relatively low cutoff of 0.0004%. Overall, results on the mock communities showed that Deblur tended to call the least amount of ASVs/OTUs among all pipelines and open-reference OTU clustering called the most.

The relative abundances determined for each study on the mock communities were similar to each other irrespective of which pipeline processed the data (Fig. 2). This finding suggests that biological conclusions based on microbial relative abundance data should be unaffected by the choice of denoising pipeline. One trend that was noticed

in the relative abundance data was that UNOISE3 tended to call higher abundances of non-reference ASVs/OTUs in each mock community except for the Extreme community where open-reference OTU clustering found the highest abundance of non-reference hits. Interestingly, the lowest identity match for any of these ASVs called by UNOISE3 in both the Zymomock and HMP mock communities was still found to be above 90% identity to the expected sequences (Figs. S1, S3) and was classified as Gammaproteobacteria by the RDP classifier using a 70% confidence threshold, suggesting it is a 16S rRNA sequence that may have been introduced by contamination, sequencing bleed-through or acquired an error early on in PCR amplification.

The relative abundances determined within the Zymomock and fungal communities were highly similar between pipelines, but markedly differed from the expected result. This finding suggests that either the expected abundances of sequences from these communities may be incorrect or all four pipelines are similarly biased. This non-agreement could also be due to steps during the sequencing processes such as PCR amplification, which may be causing primer bias (*Aird et al., 2011*) or the inclusion of contaminant organisms. In the case of the fungal community, it is possible that none of these approaches work well with ITS1 data which are more variable than 16S data. Additional fungal mock communities should be analyzed in the future to better explore this issue.

Benchmarking relative abundance profiles from different pipelines with mock communities can be useful, however, they tend to lack the diversity that is found in many real sample datasets. To address this issue, we compared resulting microbial compositions from each pipeline across three real datasets (exercise, human-associated, and soil). Weighted UniFrac, unweighted UniFrac and Bray-Curtis dissimilarity distances between the same biological samples for each approach were examined. In both cases the weighted UniFrac and Bray-Curtis intra-sample distances between all pipelines for all three datasets were small (less than a median of 0.21) (Fig. 3, Fig. S5). This complemented our previous results, showing that each pipeline had comparable microbial compositions for the mock communities. Furthermore, plotting the samples on a PCoA or NMDS resulted in the same biological samples from each pipeline grouping together (Fig. 3, Fig. S5). This indicated that a similar plot would be observed whether the researcher was using Deblur, UNOISE3, DADA2 or open-reference OTU clustering. Interestingly, Deblur did not agree with DADA2 or UNOISE3 as much as they agreed with each other on multiple occasions, based on Bray Curtis dissimilarity at the genus level (Figs. 5SA, S5B). This result is interesting, as one of the main differences between Deblur and the other two denoising pipelines is its positive filtering feature, and so we expected this feature to be driving this difference. However, when we compared the other three approaches to Deblur and Deblur without positive filtering, we found no difference (Fig. S7). Due to the similar weighted UniFrac results (Fig. 3) between the denoising pipelines, we believe that this difference is most likely due to highly similar sequences being classified into slightly different genera.

Among all of the denoising pipelines we found that Deblur agreed the least with the OTU picking approach. In most cases, communities processed using the OTU clustering approach had similar intra-sample distances between pipelines in both weighted UniFrac and Bray Curtis dissimilarity as denoising pipelines had amongst each other. The relatively

small differences seen in the abundance-based metrics indicates that choice of pipeline would have minimal impact when looking at weighted beta-diversity metrics. Although, due to denoising pipelines capability of single nucleotide resolution, they may provide more strain information than OTU clustering at 97% sequence identity would.

Unweighted UniFrac beta-diversity metrics were highly variable between pipelines indicating different bias between pipelines when determining low abundance sequences (Figs. 6A–6C). When pipelines were plotted together on a PCoA plot we found that the samples were separated by approach, rather than by sample, indicating that interpretation from unweighted UniFrac data would most likely be impacted based on the pipeline chosen to process any set of given data. We wish to highlight, however, that recent evidence has emerged that the unweighted version of UniFrac analysis can give misleading results (*Wong, Wu & Gloor, 2016*), therefore these patterns should be interpreted with caution.

To follow up on the vastly different unweighted UniFrac profiles of samples, we looked at the alpha-diversity between the same samples run by each pipeline. We found that DADA2 called the most ASVs among all denoising approaches, but overall open-reference OTU clustering found the most (Fig. 4). This was not surprising, as it agreed with the analysis of the mock communities. Interestingly, despite DADA2 finding the most ASVs overall for all three denoising pipelines, it found the least amount of ASVs per sample (Figs. S8–S10). We suspect this is due to DADA2's ability to create pooled error profiles and then pick ASVs sample by sample. Overall, open-reference OTU clustering found more OTUs per sample than ASVs found by any denoising pipelines. This indicated that the number of different organisms found within a sample is directly impacted by the choice of processing pipeline, emphasizing the difficulty in determining the true number of different organisms a sample contains. It should be noted that all of the denoising pipelines provide parameters that could be altered to increase or decrease sensitivity in identifying rare/spurious ASVs depending on a user's targeted application. One alarming result we found was the poor correlation between the number of observed OTUs/ASVs found by DADA2 and both open-reference OTU clustering and UNOISE3 (Fig. 4). This is concerning as major differences in biological signal would be seen depending on the approach that was chosen to process the data (i.e., a sample could have relatively low numbers of ASVs based on DADA2 analysis, but have relatively high numbers of ASVs/OTUs based on UNOISE3 or an OTU analysis). This issue highlights that the approaches are all very good at identifying highly abundant sequences but vary when identifying low-abundance sequences which will impact metrics that do not take into account the abundance of OTUs/ASVs.

A major difference between the three denoising pipelines was their computational run time. UNOISE3 was magnitudes faster than both DADA2 and Deblur. This is most likely due to both the programming language that UNOISE3 is implemented in (C++), as well as its simple one-pass clustering strategy. DADA2 was the slowest pipeline and, although computation time could be inconvenient for those with limited computational power, it did not reach times that were impractical even when running almost 2 million total reads. Memory usage for each program also did not reach impractical amounts when running close to 2 million reads, with DADA2 using a maximum amount of 4,071 Mb of memory which is a reasonable amount for modern computers. Memory usage by UNOISE3 did not

come close to reaching the 4 Gb memory cap on the 32-bit version, even after running 103 samples at a total read depth of 2 million reads. This suggests that for most moderately sized 16S datasets the 32bit version of UNOISE3 should be sufficient.

## CONCLUSION

In conclusion, all four pipelines are comparable when looking at weighted results that are based on the relative abundances of ASVs/OTUs while using default settings. However, the approaches do vary when looking at the number of ASVs/OTUs found and unweighted metrics such as unweighted UniFrac, while using the default settings for each denoising pipeline. The number of ASVs called did not differ between denoising pipelines in a consistent way across mock communities, suggesting that determining species richness within low-diverse samples could be problematic. However, we did find that open-reference OTU clustering consistently called more OTUs than ASV-calling pipelines. Analysis of the real datasets showed that DADA2 consistently called more ASVs than the other two denoising pipelines and that, again, open-reference OTU clustering called the most overall. More importantly, in the soil dataset and in the Extreme dataset DADA2 and the open-reference OTU clustering pipeline were capable of finding more low-abundance organisms, but DADA2 could do this without the cost of significantly increasing the number of non-reference hits. The most alarming result was the poor correlation in the number of ASVs/OTUs per sample found between DADA2 and UNOISE3 or open-reference OTU clustering. From this, we believe that choice of approach will have large impacts on determining the alpha-diversity between different samples. Looking at computational run time, we found that DADA2 was by far the slowest denoising pipeline, whereas UNOISE3 was the fastest, processing datasets more than 1200 times faster than DADA2. In the end, the choice of approach did not play a large role in weighted analyses based on microbial abundances, but did have implications on unweighted results and alpha-diversity metrics. We believe this is promising, as it indicates that no matter the choice of approach, a similar weighted biological signal will be seen. On the other hand, extreme caution is required when looking at unweighted results and alpha-diversity metrics between different pipelines.

## ACKNOWLEDGEMENTS

We would like to thank members of the Langille lab for providing feedback and suggestions for additional analysis during lab meetings. We would especially like to thank Karl Leuschen for providing previous data on a comparison between QIIME OTU clustering and DADA2 denoising.

### Funding

Jacob T. Nearing is a trainee in the Cancer Research Training Program of the Beatrice Hunter Cancer Research Institute, with funds provided by the Terry Fox Research Institute (TFRI). Gavin M. Douglas is supported by an NSERC Alexander Graham Bell Canada

Graduate Scholarship. Morgan G.I. Langille is supported by an NSERC Discovery Grant. The funders had no role in study design, data collection and analysis, decision to publish, or preparation of the manuscript.

### Grant Disclosures
The following grant information was disclosed by the authors:
Terry Fox Research Institute (TFRI).
NSERC Alexander Graham Bell Canada Graduate Scholarship.
NSERC Discovery Grant.

### Competing Interests
The authors declare there are no competing interests.

### Author Contributions
- Jacob T. Nearing conceived and designed the experiments, performed the experiments, analyzed the data, prepared figures and/or tables, authored or reviewed drafts of the paper, approved the final draft.
- Gavin M. Douglas conceived and designed the experiments, analyzed the data, authored or reviewed drafts of the paper, approved the final draft.
- André M. Comeau contributed reagents/materials/analysis tools, authored or reviewed drafts of the paper, approved the final draft.
- Morgan G.I Langille conceived and designed the experiments, authored or reviewed drafts of the paper, approved the final draft.

### DNA Deposition
The following information was supplied regarding the deposition of DNA sequences:
The Human Microbiome Project mock community and Zymomock community sequences described here are accessible via ENA accession number PRJEB24409.

### Data Availability
Scripts that were used to run all data analysis can be found at: https://github.com/nearinj/Denoiser-Comparison.

### Supplemental Information
Supplemental information for this article can be found online at http://dx.doi.org/10.7717/peerj.5364#supplemental-information.

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
