# Peer review of "Denoising the Denoisers: an independent evaluation of microbiome sequence error-correction approaches"

_PeerJ, doi:10.7717/peerj.5364_

## Round 0.1 · original submission · Major Revisions

All the reviewers found the work interesting, timely and important and should be encouraged. The evaluation was also very comprehensive. However, they raised some concerns about the comparability of methods evaluated as well as the reproducibility and interpretability of the results. These concerns could be addressed by providing more clarification or analyses.

I would recommend also comparing the ASV-based methods to a traditional de novo OTU-based method on a real dataset. Various correlation measures, such as Pearson correlation for alpha-diversity measures, Mantel correlation for beta-diversity measures, and rank correlation for genus abundances, could be used for evaluation. The results will answer the question whether the OTU-based methods could obtain statistically similar results as the state-of-the-art ASV-based methods. If so, OTU-based methods are still not outdated.

Please add unweighted UniFrac distance in the comparison. Unweighted and weighted UniFrac provide very different views of the microbial community.

Reviewer 1 ·

Basic reporting

This manuscript uses clear and professional English throughout. They also cite appropriate references from the literature and provide sufficient background. Tables and figures are well organized and informative. They hypotheses presented were supported with relevant results.

Experimental design

The research question is well defined. The authors' primary goal is to assess the computational time and accuracy of different ASV calling methods. This question is highly relevant to the field.

Please clarify why an abundance filter of .1% was selected, and why the rarefaction levels presented were selected. Was there an obvious cut point?

QIIME1 is no longer supported, so I wonder why the authors chose to use this method instead of several other publicly available packages to calculate diversity and generate principle components.

I would be interested in seeing comparisons with Deblur without the positive filtering method. It seems likely that this positive filtering may bias findings towards sequences present in greengages.

Validity of the findings

The conclusion of the paper seems rather weak. From reading this paper, I did not come to the same conclusions as the authors; it seemed to me that based on this data, the three methods were not comparable. The authors make this statement based on weighted Unifrac differences, though this alone cannot justify this statement as different experimental questions will justify the use of different packages.

The authors often state that DADA2 calls more ASVs than other methods, but does not venture into whether these are spurious calls or likely to be true. This can be best evaluated with the mock communities where the "answer" is known. Table S2 appears to be the only mock community with discordant identification of ASVs between methods, and DADA2 identifies more ASVs correctly, though it is unclear whether this is just a function of identifying more total ASVs.

The authors should clarify the implications of using a pooled verses sample-by-sample workflow. While the sample-by-sample approach may be more computationally efficient, it looses the ability to correct errors within batches. For multi-run experiments, this could lead to even greater batch effects.

Is it a fair comparison to use dada2 filtering scripts before inputting data into all three pipelines. Would the typically user of these pipelines do this filtering since other parameters were left at default for this comparison?

Additional comments

The authors evaluate three ASV calling methods on multiple mock communities and datasets. Overall, this is a solid manuscript that makes a significant contribution to the literature.

Reviewer 2 ·

Basic reporting

No comment

Experimental design

No comment

Validity of the findings

No comment

Additional comments

Identification of ASVs or zOTUs from microbe amplicon sequencing is a raising area. Most of the proposed methods are not well benchmarked yet and a fair comparison between these methods is in great need. Hence the efforts and findings of the paper should be encouraged. However, the following issues in the experiment design should be further considered in order to achieve a solid benchmark:
1) It should be noted that the methods tested in the paper used different chimera-checking pipelines (and possibly also other pre-processing steps), this would affect the initial point of the evaluations. Please check the consistency of chimera-checked sequences generated from different pipelines.
2) For denoised sequences, a 97% similarity is quite a loose match which is at least unintended for the denoising algorithms. I suggest to draw a complete similarity-proportion curve for each method (at least on one example) instead of using the 97% threshold alone.
3) For the claim "DADA2 tended to find more ASVs than the other two methods", it should be noted that each algorithm provided parameter settings that can trade-off between the recall and the quality of ASVs (e.g., for unoise you can tune the lower limit of sequence count for detecting an ASV) . It is fair to plot the complete trade-off curve using different parameter settings before making the claim.
4) The proportion of unexpected but matched ASVs is large, this should be explained.

Reviewer 3 ·

Basic reporting

This study, presented by Nearing et al., is an important addition to the scientific literature. With the plethora of tools and software becoming available for 16S rRNA gene sequence processing, an unbiased assestment of ASV processing tools is imperative to the microbiome field. This reviewer is also encouraged to see that testing communities were pulled from human and soil microbiomes as it is important to test the performance of these software on a variety of bacterial communities containing many different types of species and levels of diversity.
In multiple places thoroughout the manuscript, this reviewer required more detail as to methods, results, and interpretation, and in some places found differing accounts of the results between the manuscript and figure legends. Specific lines with suggestions are outlined below. However, a lot of these questions and concerns could have been mitigated by including the specific scripts/code for the processing and analysis presented here. Because this is a manuscript which focusses on the comparison of various software, the exact commands that have been used to generate the BIOM tables should be provided. I do appreciate that most of the processing has been completed with Microbiome Helper (which is open source and public); however, steps to reproduce, for example, the UNOISE3 Pipeline (lines 118-126) would be difficult given the current information. Further, reproducing the analyses within would be extremely difficult without the appropriate code to do so. This supplemental material would be advantageous to the community as more tools become available, or if readers are interested in testing these 3 tools against their own mock communities.

Experimental design

In general, the experimental design is sound. As mentioned above, having the processing pipeline available in the supplemental material would aid in the reproducibly and visibility of this work.
Through the Materials and Methods the authors refer to "soil", "mouse", and "human" datasets; it would be advantageous to orient the reader in the Sequence Acquisition section as to which of these (HMP mock community, Zymomock, Extreme, fungal) are from which environments, and to use consistent naming throughout the text.
line 85: The HMP produced 2 mock communities, both with the same species composition but with different relative abundances. It would be advantageous to indicate which of these is used here without having to refer to Corneau, Douglas, & Langille 2017.
line 102: Why were cutoffs of 270 and 210 base pairs employed? Please detail here.
line 105: This sentence is repetitive of that above it.
line 148: Which database was BLASTN used against?
line 154: Is this 0.1% filter in addition to analyses done with a 0% filter? Wording is not clear here. (This does become apparent in the Results but I would still request that it is clarified here.)
line 173: It is not clear here what is meant by "medium stringencies" here.

Validity of the findings

I disagree with the authors indication that the biological results from these methods are unaffected by the chosen ASV processing tool. Having significantly different levels of ASVs assigned from each method (as shown in Fig1) indicates that these tools are identifying a different number of amplicon sequences in these samples. Although the genus-level assignments are very similar, differing numbers of ASVs are likely to severely impact alpha and beta-diversity metrics when performed at the ASV-level; the implications of these differences have been previously shown at the OTU level (for e.g. DOI 10.1186/s40168-017-0314-2). Further, it is more common to calculate these metrics at the OTU/ASV-level than at the genus-level (for e.g. as done in these recent publications doi: 10.1128/AEM.02335-17,https://doi.org/10.1073/pnas.1717617115), and is actually suggested in the DADA2 Pipeline Tutorial (1.6). I do understand that calculating beta-diversity at the genus-level is useful here in order to identity that these 3 methods all produce similar genus-level identifications of this community. However, it is imperative to point out that most investigations of the microbiome calculate beta diversity at the OTU or ASV level, not based on genus-level assignments. A supplemental to Fig3 should indicate whether similar analyses at the ASV-level produce similar results, and a point in the Discussion be made as to why this is important.
As a follow up to this point, it would be interesting to note the differential in the number of ASVs called by each method in the processing of the real soil dataset.
Fig1: How was the number of expected ASVs per community identified? Details on how this was determined should be provided in the Materials and Methods. This is a cruical detail which determines how the results of this Figure are interpreted.
Fig1: Due to a lack of SupFig legends (see below) I can't be sure which is missing, but why is one dataset missing from SupFig1?
SupTable1-4: It is not clear to me whether the "Percent Abundance" column is the Expected or Observed relative abundance of the organism in the mock community. Regardless, it would be advantageous to include both the Expected and Observed abundances of each organism here in order to make an informed descion as to tool performance. The text also indicates that information as to percent identity (100% match, >97% match) will be found in these Tables; however, they are not present.
line 206: Sequencing depths of the mock (and real) communities should be included in the Materials and Methods in order for the reader to estimate whether low abundance taxa were missed due to low sequencing depth.
Fig2: Unique colours should be used to distinguish taxa.
-lines288-295: These results should be included in the manuscript.
I really appreciate the inclusion of Supp Fig 6; the distribution of samples is often ignored and this reviewer appreciated being able to take a peak at them via this Figure.

Additional comments

I was unable to find the Figure legends for any of the Supplemental Figures. This may be an issue with file upload, or my own naivity, but it still should be noted.
line 50: Clarity is needed here; 16S rRNA gene sequencing is so widely adopted because its sequence tends to be similar among members of a species but differ across species. Also, because of its structure it uniquely allows areas of high conservation to be used in the creation of (near) universal primers which can be used to target areas of high variation. The way in which the authors currently introduce 16S rRNA gene sequencing here doesn't give the methodology justice.
line 54: Many research groups and projects still rely on OTUs, they are not yet "in the past". For example, the following studies were published in 2018: https://doi.org/10.1186/s40168-017-0389-9, doi:10.1038/s41598-018-23261-1, doi: 10.1128/AEM.02335-17.
line 61: There seem to be missing symbols following many references within the test which are appearing as empty boxes.
SupTable1-4: Spelling error in Table headers.
-The text indicates that Fig3a is based on genus-level distances; the Figure legend indicates that they are based on ASV-level distances.
-SupFig7: need y-axis labels.
-Table 1 is only mentioned in the Discussion; the information contained within would be more beneficial if it was brought to the readers attention at the beginning of the Results section.
-line335: This should be mentioned in the Materials and Methods.
-line357: What is the sequence bleed-through rate of an Illumina HiSeq? This information would be helpful here to make the comparison clearer.

---

## Round 0.2 · Minor Revisions

The revised version improved significantly over the original version. However, one reviewer still had serious concerns about the comparability of these methods. I would recommend revising the text to reflect the fact that the evaluation is comparing the entire "pipelines" instead of "error correction methods". It is also informative to address the limitations of the evaluation in the discussion.

Reviewer 1 ·

Basic reporting

no comment

Experimental design

no comment

Validity of the findings

no comment

Additional comments

All of my comments have been addressed.

Reviewer 2 ·

Basic reporting

no comment

Experimental design

no comment

Validity of the findings

no comment

Additional comments

The revision is completely unsatisfactory in my point of view. See below.
1) ”but the chimera-checking methods employed by each method are tightly integrated within the documented use of each denoising method and it is unlikely that a typical user would choose a different chimera checking method from a different bioinformatic pipeline" -- This is not a reason for not carrying out additional experiments. A capable evaluation paper is not just about running code and collecting results. The authors should try to find the cause that make the discrepancy and suggest potential directions for improving. For the chimera-checking issue, what the authors should do is :(1) Create a pre-processed, chimera-free data set that would not be further filtered by the tested algorithms (or by-pass additional fitering steps of the tested algorithms if possible), and compare the observed outcome across algorithms with the one using raw data. (2) If chimera-checking does affect the ASV results, try to explain the cause.
2) "however, the majority of users will most likely use the default values and for this reason we tested these default parameters only" -- Again, this not a reason for not taking comparisons. Parameter varation analyses and trade-off analyses are basic requirements for an evalutaion paper.
3) “We also found a high proportion of matches to a database but not the expected sequences on multiple occasions. We believe this is most likely due to the introduction of contaminate DNA sequences during the preparation of the mock communities for sequencing.” -- This is a completely baseless guess, because inproper choice of database query assignment criteria or representative sequences would as well lead to the case. If should be further noted that different methods report quite different portions of unexpected sequences, this cannot be explained by contamination. Moreover, large-scale contamination is not expected to happen for a mock community study. If unfortunately (though dubiously) this happened to be true, the authors would have to use other less-contaminated data for test instead.

Reviewer 3 ·

Basic reporting

All previous comments have been adequately addressed.

Experimental design

No comment.

Validity of the findings

No comment.

---

## Round 0.3 · accepted · Accept

After the additional clarification and acknowledgment of limitations, the work is now suitable for publication. I believe the work is important, informative and timely, and is a nice contribution to the growing field.

#